

# Isolation and identification of L-asparaginase-producing endophytic fungi from the Asteraceae family plant species of Iran

Sareh Hatamzadeh[1], Kamran Rahnama[1], Saeed Nasrollahnejad[1], Khalil Berdi Fotouhifar[2], Khodayar Hemmati[3], James F. White[4] and Fakhtak Taliei[5]

[1] Department of plant protection, Faculty of plant production, Gorgan University Of Agricultural Sciences And Natural Resources, Gorgan, Iran
[2] Department of plant protection, Faculty of Agricultural Sciences and Natural Resources, University of Tehran, Tehran, Iran
[3] Department of Horticulture, Faculty of Plant Production, Gorgan University Of Agricultural Sciences And Natural Resources, Gorgan, Iran
[4] Department of Plant Biology, Rutgers University, New Brunswick, NJ, United States of America
[5] Department of Plant Production, Gonbad Kavous University, Gonbad Kavous, Iran

Corresponding author
Kamran Rahnama,
Rahnama@gau.ac.ir,
kamranrahnama1995@gmail.com

## ABSTRACT

L-asparaginase is an important anticancer enzyme that is used in the first line treatment of acute lymphoblastic leukemia. This study was conducted to isolate L-asparaginase-producing endophytic fungi from medicinal plants of family Asteraceae. Seven healthy medicinal plants from family Asteraceae were selected for the isolation of endophytic fungi using standard surface sterilization techniques. A total of 837 isolates belonging to 84 species were comprised of the stem (55.6%), leaf (31.1%), root (10.6%) and flower (2.7%). Initial screening of L-asparaginase-producing endophytes was performed by qualitative plate assay on modified Czapex dox's agar medium. L-asparaginase activity of fungal endophytes was quantified by the nesslerization method. Identification of endophytic fungi was performed using both morphological characteristics and phylogenetic analyses of DNA sequence data including ribosomal DNA regions of ITS (Internal transcribed spacer) and LSU (partial large subunit rDNA), TEF1 (Translation Elongation Factor) and TUB ($\beta$-tubulin). Of the 84 isolates, 38 were able to produce L-asparaginase and their L-asparaginase activities were between 0.019 and 0.492 unit/mL with *Fusarium proliferatum* being the most potent. L-asparaginase-producing endophytes were identified as species of *Plectosphaerella*, *Fusarium*, *Stemphylium*, *Septoria*, *Alternaria*, *Didymella*, *Phoma*, *Chaetosphaeronema*, *Sarocladium*, *Nemania*, *Epicoccum*, *Ulocladium* and *Cladosporium*. This study showed that endophytic fungi from Asteraceae members have a high L-asparaginase-producing potential and they can be used as an alternative source for production of anticancer enzymes.

## INTRODUCTION

Endophytes are microorganisms that reside inside plant tissues without causing any symptoms or obvious negative effects to the host plants (*Gond et al., 2012*). Endophytic fungi from medicinal plants have been considered to be a rich source of novel natural products for medical and commercial exploitation (*Gutierrez, Gonzalez & Ramirez, 2012*). The close symbiotic relationship between endophytic fungi and host plants gives endophytes a potent ability to produce novel bioactive compounds whose production is fueled by host plant carbohydrates (*Aly, Debbab & Proksch, 2011*). These bioactive compounds increase plant resistance to pathogens and herbivores, enhanced competitive abilities and enhanced growth (*Zhang, Song & Tan, 2006*). Endophytic fungal bioactive metabolites may be useful as novel drugs due to their wide variety of biological activities (*Guo et al., 2008*). In recent years, endophytic fungi have been viewed as a source of secondary metabolites, including anticancer, anti-inflammatory, antibiotic and antioxidant agents (*Guo et al., 2008*; *Bungihan et al., 2011*; *Debbab et al., 2009*; *Gutierrez, Gonzalez & Ramirez, 2012*; *Strobel et al., 2004*).

Enzymes produced by microorganisms are used for medical and industrial purposes. L-asparaginase is one such enzyme that hydrolyzes asparagine to aspartic acid and ammonia (*Patil, Patil & Mahjeshwari, 2012*). L-asparaginase enzymes in the food industry are used as an admixture to reduce the acrylamide produced by the high temperature in starchy foods and reduce the risk of cancer (*Xu, OrunaConcha & Elmore, 2016*). This enzyme is one of the most important biochemical therapeutic enzymes used in the treatment of various types of leukemia, such as acute lymphoblastic leukemia in children (*McCredie & Ho, 1973*; *Patil, Patil & Mahjeshwari, 2012*). In cancer treatment, L-asparaginase removes L-asparagine in the serum, depriving tumor cells of the large amounts of asparagine required for growth (*Asthana & Azmi, 2003*). Currently, L-asparaginase derived from *Escherichia coli* is the main source of L- asparaginase (*Batool et al., 2015*). However, side effects of this enzyme derived from bacteria include chills, fever, abdominal cramps and fatal hyperthermia (*Hosamani & Kaliwal, 2011*). L-asparaginase derived from eukaryotes may induce relatively less toxicity and reduced immune response (*Asthana & Azmi, 2003*). Considering the importance of L-asparaginase in the treatment of leukemia, finding new sources of this enzyme that can produce high levels of enzyme with minimum side effects is a priority (*Theantana, Hyde & Lumyong, 2009*). Microorganisms such as fungi have the ability to produce extracellular enzymes in high quantities, which are easily extracted and purified, may provide ideal sources of L-asparaginase. In addition, fungal species due to their eukaryotic nature may have enzymes more comparable to human enzymes that may be used in treatment of cancer with better success than enzymes of other microorganisms (*Serquis & Oliveira, 2004*). L-asparaginase from endophytic fungi isolated from medicinal plants has been reported in recent years (*Theantana, Hyde & Lumyong, 2007*).

The appropriate selection of host plants is important to increase the chances of isolation of novel endophytes which may produce new bioactive metabolites (*Ratklao, 2013*). In this study, seven Iranian medicinal plants, including: *Matricaria chamomilla, Matricaria parthenium, Athemis triumfetii, Anthemis altissima, Achillea millefolium,*

*Achillea filipendulina* and *Cichorium intybus,* were selected for the isolation of fungal endophytes and screening them for L-asparaginase activity based on use in traditional medicine, habitat, and diversity of these plant species in the Golestan province of Iran.

## MATERIALS & METHODS

### Isolation and identification of fungal endophytes

From May until September 2015, the plant specimen (stem, root, leaf and flower) were obtained from seven healthy medicinal plants: *Matricaria chamomilla, Matricaria parthenium, Anthemis triumfetii,, Anthemis altissima, Achillea millefolium, Achillea filipendulina* and *Cichorium intybus* growing in the natural ecosystem in Northeastern of Iran. The samples were stored in polyethylene bags at 4 °C (*Waksman, 1916*). Samples were washed thoroughly in distilled water. The Surface sterilization was performed by sequential immersion of samples in sodium hypochlorite (3 times in 5% NaOCl for 3–8 min each, depending on the type of samples) followed by 75% ethanol for 1 min and rinsed five times in sterile distilled water. The samples were dried on sterile blotters under the laminar air flow. The surface-sterilized samples were cut into about $0.5 \times 1$ cm$^2$ using a sterile scalpel. 200 segments from stem, leaf, root and flower (four segments per Petri plate) for each plant species were placed equidistantly on Potato Dextrose Agar medium (PDA, Merck, Darmstadt, Germany) supplemented with tetracycline (50 mg/L) to inhibit bacterial growth. Three replicates of Petri dishes were used per plant sample. The petri plates were incubated at $28 \pm 2$ °C with 12 h light and dark cycles for up to 6 to 8 weeks. To verify sterility controls were prepared by spreading 100 µL aliquots of the water from final rinse solutions onto PDA medium plates and incubated for 2 weeks at $28 \pm 2$ °C with 12 h light and dark cycles. The absence of fungal growth in controls indicated effective sterilization, while mycelial growth from plant samples was indicative of endophyte isolation. Colonies that emerged from tissue segments were picked up and transferred to antibiotic-free PDA to enable identification. Colonization Frequency (CF) of endophytes was calculated as described by *Khan et al. (2010)*.

$$\text{Colonization Frequency of Fungi (\%)} = \frac{\text{Number of Isolates of Taxon from Each Segment}}{\text{Total Number of Segments}} \times 100$$

Identification was achieved using morphological and molecular methods. Morphological identification of isolates was performed based on the fungal colony morphology, characteristics of the spores and reproductive structures using standard identification manuals (*Barnett & Hunter, 1999*; *Bensch et al., 2012*; *Boerema et al., 2004*; *Simmons, 2007*; *Booth, 1971*).

### Molecular identification of endophytic fungi

Endophytic fungal isolates were grown in 200 mL of potato dextrose broth (PDB) for 7 days at 28 °C. The mycelia were washed with distilled water and ground with liquid nitrogen. The nucleic acid was extracted using the cetyl trimethyl ammonium bromide (CTAB) method (*Dayle et al., 2001*). Strains were sequenced with four: molecular markers, including ITS (Internal transcribed spacer), LSU (partial large subunit nrDNA), TEF1

**Table 1  Primer combinations used for molecular identification.**

| Locus | Primer | Primer sequence 5′ to 3′: | Orientation | Reference |
|---|---|---|---|---|
| TEF-1$\alpha$ | EF1-983F | GCCYGGHCAYCGTGAYTTYAT | Forward | *Rehner & Buckley (2005)* |
| | Efgr | GCAATGTGGGCRGTRTGRCARTC | Reverse | *Rehner & Buckley (2005)* |
| $\beta$-tubulin | T1 | AACATGCGTGAGATTGTAAGT | Forward | *O'Donnell & Cigelnik (1997)* |
| | $\beta$-Sandy-R | GCRCGNGGVACRTACTTGTT | Reverse | *O'Donnell & Cigelnik (1997)* |
| LSU | LROR | CC CGC TGA ACT TAA GC | Forward | *Vilgalys & Hester (1990)* |
| | LR5 | TCCTGAGGGAA ACTTCG | Reverse | *Vilgalys & Hester (1990)* |
| ITS | ITS5 | GGAAGTAAAAGTCGTAACAAGG | Forward | *White et al. (1990)* |
| | ITS4 | TCCTCCGCTTATTGATATGC | Reverse | *White et al. (1990)* |

(Translation elongation factor) and TUB ($\beta$-tubulin) using primer sets listed in Table 1. PCR amplifications were performed on a GeneAmp PCR System 9600 (Perkin Elmer, USA) in a total volume of 12.5 µL solution containing 10–20 ng of template DNA, 1 × PCR buffer, 0.7 µL DMSO (99.9%), 2 mM MgCl2, 0.5 µM of each primer, 25 µM of each dNTP and 1.0 U Taq DNA polymerase (NEB). The amplification process was initiated by pre-heating at 95 °C for1 min, followed by 40 cycles of denaturation at 95 °C for 30s, with primer annealing at the temperature stipulated in Table 2, extension at 72 °C for 10s, and a final extension at 72 °C for 5 min. The products of the PCR reaction were then examined by electrophoresis using 1% (w/v) agarose gel, stained with gel red (Biotium®) and visualized with a UV transilluminator (UVP MultiDoc-It™, Analytik Jena, Germany). BLAST analysis was carried out in the NCBI database. All sequences were deposited in NCBI's GenBank Database.

## Screening of L-asparaginase-producing endophytes

The isolated endophytic fungi were screened for their ability to produce asparaginase. Mycelial plug was inoculated onto Modified Czapex Dox (McDox) agar [agar powder (20.0 g/ L$^{-1}$), glucose (2.0 g/L$^{-1}$), L-asparagine (10.0 g/L$^{-1}$), KH$_2$PO$_4$ (1.52 g/L$^{-1}$), KCl (0.52 g/L$^{-1}$), MgSO$_4$·7H$_2$O (0.52 g/L$^{-1}$), CuNO$_3$·3H$_2$O (0.001g/L$^{-1}$), ZnSO$_4$·7H$_2$O (0.001 g/L$^{-1}$), FeSO4·7H2O (0.001 g/L$^{-1}$)], l-asparagine (10.0 g/L$^{-1}$) and 0.3 mL of 2.5% phenol red dye (indicator). Controls were prepared by inoculating mycelial plugs on Czapex Dox agar without asparagine. Triplicates for each isolate were prepared. All Petri plates were incubated at 26 ± 2 °C. After 5 days of incubation, the diameter of the pink zone was evaluated (*Gulati, Saxena & Gupta, 1997*).

## Estimation of L-asparaginase activity

The L-asparaginase positive fungal isolates were inoculated using 5 mm fungal mycelial plugs into 200 mL of McDox broth and incubated for 5 days at 36 ± 2 °C and 120 rpm. L-asparaginase was estimated by Nesslerization as described by *Imada et al. (1973)*. After incubation, 100 µl of broth (crude enzyme) was pipetted into 2 ml tubes. After that, 100 µl of Tris HCl (pH 7), 200 µl of 0.04 M asparagine and 100 µl of sterile distilled water (SDW) were added. The mixture was incubated at 37 ± 2 °C for 1 h. After incubation, 100 µl of 1.5 M Trichloroacetic Acid (TCA) was then added to stop the enzymatic reaction.

**Table 2 Colonization frequency of endophytic fungi.** Two hundred segments of each sample were plated for frequency analysis. ($n = 10$) Not detected: –.

| Endophytic Fungi | Host Plant | Colonization frequency | | | | Total |
|---|---|---|---|---|---|---|
| | | Stem | Leaf | Root | Flower | |
| *Fusarium redolens* | *Achillea millefolium* | 5.5 | – | – | – | 11 |
| *Septoria saposhnikoviae* | *A. millefolium* | 1 | – | – | – | 2 |
| *Paraophiobolus arundinis* | *A. millefolium* | 4.5 | – | – | – | 9 |
| *Stemphylium amaranthi* | *A. millefolium* | 2.5 | – | – | – | 5 |
| *Cladosporium ramotenellum* | *A. millefolium* | – | 5 | – | – | 10 |
| *Septoria tormentillae* | *A. millefolium* | 1.5 | – | – | – | 3 |
| *Septoria lycopersici var. lycopersici* | *A. millefolium* | 1 | – | – | – | 2 |
| *Septoria sp.* | *A. millefolium* | 2.5 | – | – | – | 5 |
| *Fusarium oxysporum* | *A. millefolium* | 5 | – | – | – | 10 |
| *Septoria malagutii* | *A. millefolium* | 1.5 | – | – | – | 3 |
| *Fusarium sp.* | *A. millefolium* | – | 5.5 | – | – | 11 |
| *S. tormentillae* | *A. millefolium* | 3.5 | – | – | – | 7 |
| *Alternaria infectoria* | *A. millefolium* | 5.5 | – | – | – | 11 |
| *Leptosphaerulina saccharicola* | *A. millefolium* | 2.5 | – | – | – | 5 |
| *Alternaria burnsii* | *A. millefolium* | – | 4.5 | – | – | 9 |
| *Alternaria sp.* | *A. millefolium* | – | 5.5 | – | – | 11 |
| *Nemania serpens* | *A. millefolium* | 3.5 | – | – | – | 7 |
| *Stemphylium vesicarium* | *A. millefolium* | 4 | – | – | – | 8 |
| *Fusarium avenaceum* | *A. millefolium* | – | – | 6 | – | 12 |
| *Fusarium sp.* | *A. millefolium* | – | 6 | – | – | 12 |
| *Paraphoma chrysanthemicola* | *A. millefolium* | – | 5 | – | – | 10 |
| *F. oxysporum* | *Achillea filipendulina* | 8 | – | – | – | 16 |
| *Fusarium sp.* | *A. filipendulina* | 5.5 | – | – | – | 11 |
| *Preussia africana* | *A. filipendulina* | 1.5 | – | – | – | 3 |
| *Plectosphaerella cucumerina* | *A. filipendulina* | 6.5 | – | – | – | 13 |
| *Antennariella placitae* | *A. filipendulina* | 5 | – | – | – | 10 |
| *Fusarium acuminatum* | *A. filipendulina* | – | 6.5 | – | – | 13 |
| *Acremonium sclerotigenum* | *A. filipendulina* | – | 6 | – | – | 12 |
| *Colletotrichum tanaceti* | *A. filipendulina* | 4.5 | – | – | – | 9 |
| *Trametes versicolor* | *A. filipendulina* | 1.5 | – | – | – | 3 |
| *A. burnsii* | *Anthemis altissima* | 4 | – | – | – | 8 |
| *Lewia infectoria* | *A. altissima* | – | – | 8 | – | 16 |
| *P. chrysanthemicola* | *A. altissima* | 6 | – | – | – | 12 |
| *Aspergillus calidoustus* | *A. altissima* | – | 5.5 | – | – | 11 |
| *Bjerkandera adusta* | *A. altissima* | 2.5 | – | – | – | 5 |
| *Schizophyllum commune* | *A. altissima* | 3.5 | – | – | – | 7 |
| *A. infectoria* | *A. altissima* | 3 | – | 1.5 | – | 9 |
| *Paraphoma sp.* | *A. altissima* | 4.5 | 1 | – | – | 11 |
| *F. acuminatum* | *A. altissima* | 8 | – | – | | 16 |

*(continued on next page)*

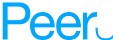

| Endophytic Fungi | Host Plant | Colonization frequency | | | | Total |
|---|---|---|---|---|---|---|
| | | Stem | Leaf | Root | Flower | |
| *Stemphylium botryosum* | *A. altissima* | – | 6.5 | – | – | 13 |
| *N. serpens* | *A. altissima* | 4 | – | – | – | 8 |
| *Fusarium proliferatum* | *A. altissima* | 8.5 | – | – | – | 17 |
| *Plenodomus tracheiphilus* | *A. altissima* | 4.5 | – | – | – | 9 |
| *Phoma tracheiphila* | *A. altissima* | 6 | 1.5 | – | – | 15 |
| *Ulocladium consortiale* | *A. altissima* | – | 6.5 | – | – | 13 |
| *P. cucumerina* | *A. altissima* | – | – | 6 | – | 12 |
| *Cladosporium limoniforme* | *A. altissima* | 6 | – | 6 | – | 24 |
| *Sarocladium strictum* | *A. altissima* | – | – | 5.5 | – | 11 |
| *Verticillium dahliae* | *A. altissima* | 4.5 | – | – | – | 9 |
| *F. avenaceum* | *A. altissima* | – | – | – | 5.5 | 11 |
| *Didymella tanaceti* | *A. altissima* | 2 | – | – | – | 4 |
| *Chaetosphaeronema sp.* | *Anthemis triumfetii* | – | 6 | – | – | 12 |
| *Chaetosphaeronema hispidulum* | *A. triumfetii* | – | 7 | – | – | 14 |
| *P. chrysanthemicola* | *A. triumfetii* | 6.5 | – | – | – | 13 |
| *Chaetosphaeronema achilleae* | *A. triumfetii* | 5 | 1 | – | – | 12 |
| *C. achilleae* | *A. triumfetii* | – | 4 | – | – | 8 |
| *S. amaranthi* | *A. triumfetii* | – | 7 | – | – | 14 |
| *Paraphoma sp.* | *A. triumfetii* | 7 | – | – | – | 14 |
| *Alternaria sp.* | *A. triumfetii* | 6 | 2 | – | – | 16 |
| *Alternaria sp.* | *A. triumfetii* | 7 | 2 | – | – | 18 |
| *S. vesicarium* | *Matricaria parthenium* | – | 4.5 | – | – | 9 |
| *Arthrinium phaeospermum* | *M. parthenium* | – | – | – | 1 | 2 |
| *Epicoccum nigrum* | *M. parthenium* | 4 | – | – | – | 8 |
| *Aspergillus chevalieri* | *M. parthenium* | – | 4.5 | – | – | 9 |
| *Trichaptum biforme* | *M. parthenium* | 1.5 | – | – | – | 3 |
| *Phoma haematocycla* | *Matricaria chamomilla* | 5 | 1 | – | – | 12 |
| *Paramyrothecium roridum* | *M. chamomilla* | – | – | 6.5 | – | 13 |
| *S. amaranthi* | *M. chamomilla* | – | 7 | – | – | 14 |
| *Xylariaceae sp.* | *M. chamomilla* | 6 | – | – | – | 12 |
| *E. nigrum* | *M. chamomilla* | 4 | – | – | – | 8 |
| *Cladosporium tenuissimum* | *Cichorium intybus* | – | 5.5 | – | – | 11 |
| *E. nigrum* | *C. intybus* | 3.5 | – | – | – | 7 |
| *Septoria cerastii* | *C. intybus* | 3.5 | – | – | – | 7 |
| *P. cucumerina* | *C. intybus* | – | – | 5 | – | 10 |
| *C. tanaceti* | *C. intybus* | 7.5 | – | – | – | 15 |
| *Stephanonectria keithii* | *C. intybus* | – | 2 | – | – | 4 |
| *Alternaria solani* | *C. intybus* | 6 | – | – | – | 12 |
| *B. adusta* | *C. intybus* | 3.5 | – | – | – | 7 |
| *Torula herbarum* | *C. intybus* | – | 3.5 | – | – | 7 |
| *Alternaria embellisia* | *C. intybus* | – | 5.5 | – | – | 11 |

**Table 2** (*continued*)

| Endophytic Fungi | Host Plant | Colonization frequency | | | | |
|---|---|---|---|---|---|---|
| | | **Stem** | **Leaf** | **Root** | **Flower** | **Total** |
| *Stemphylium globuliferum* | *C. intybus* | 4 | – | – | – | 8 |
| *A. sclerotigenum* | *C. intybus* | – | – | – | 5 | 10 |
| *Penicillium canescens* | *C. intybus* | – | 2 | – | – | 4 |
| *Diaporthe novem* | *C. intybus* | 9.5 | – | – | – | 19 |
| Number of isolates | | 466 | 259 | 89 | 23 | 837 |

Finally, 100 µl of the mixture was pipetted into fresh tubes containing 750 µl SDW and 300 µl of Nessler's reagent and incubated at 28 ± 2 °C for 20 min and the amount of enzyme activity was measured by determining the absorbance of samples at 450 nm using UV-Visible spectrophotometer (Jenway model 6315). One unit of asparaginase is expressed as the amount of enzyme that catalyzes the formation of 1 µmol of ammonia per minute at 37 ± 2 °C (*Theantana, Hyde & Lumyong, 2007*).

$$\text{Units/ml enzyme} = \frac{(\mu\text{mol of NH3 liberated})(0.6)}{(0.1)(60)(0.2)}$$

0.6 = Initial volume of enzyme mixture in mL
0.1 = Volume of enzyme mixture used in final reaction in mL
60 = Incubation time in minutes
0.2 = Volume of enzyme used in mL

## Statistical analysis

The final experiment was conducted using a completely randomized design with triplicates for each parameter assessed. The data were statistically analyzed using the software Statistical Package for the Social Sciences (SPSS) version 17.0. One-way ANOVA with least significant difference ($\text{LSD}_{(0.01)}$) were applied to analyze all the data collected.

## RESULTS

### Identification of fungal endophytes

Endophytes were obtained from all seven medicinal plant species with a total of 837 isolates from 200 each of leaf, stem and root segments. Endophytes were mostly recovered from *A. altissima* (241 isolates), followed by *A. millefolium* (163 isolates), *A. triumfetii* (121 isolates), *C. intybus* (132 isolates), *A. filipendulina* (90 isolates), *M. chamomilla* (59 isolates) and *M. parthenium* (31 isolates) (Table 2). Due to the large number of fungal endophytes, the isolates were further classified into 84 morphotypes based on the different morphological and cultural characteristics. Eighty-four endophytic fungal species belonging to Ascomycota and Basidiomycota were identified using morphological and molecular methods. Few endophytic fungi such as *Acremonium sclerotigenum*, *Alternaria burnsii*, *Bjerkandera adusta*, *Colletotrichum tanaceti*, *Epicoccum nigrum*, *Fusarium acuminatum*, *Paraphoma chrysanthemicola*, *Plectosphaerella cucumerina* and *Stemphylium amaranthi* showed wide distributions in the host plants and were isolated from most plants studied. Also, a higher number of endophytes were recovered from stem tissues of all seven plant

species (Table 2). The percent colonization frequency of endophytes varied in the plant parts with stem fragments harboring 55.6% of endophytic isolates followed by leaves with 31.1% and least for the isolates from flower samples. Many isolates belonged to the genera *Alternaria, Fusarium, Phoma, Chaetosphaeronema* and *Plectosphaerella* which colonized more than one plant part. The isolates of *Fusarium* were recovered from stem, leaf, flower and root while *Phoma* spp. was obtained from stem and leaf sample. Tissue specificity was also observed for some endophytes. This was most evident in the *Septoria* species that were found only in the stem tissues. Basidiomycetous endophytes such as *Trametes versicolor, Bjerkandera adusta, Trichaptum biforme* and *Schizophyllum commune* was isolated from stem tissues. *Fusarium* spp. was found as the dominant endophytes with 140 isolates, followed by *Alternaria* spp. (105 isolates). The results indicated that the species composition and frequency of endophyte species was found to be dependent on the tissue and host plant.

## Screening of L-asparaginase-producing endophytes by qualitative plate assay

All endophytic fungal isolates were screened for their ability to produce L-asparaginase by qualitative rapid plate assay. Of the eighty-four fungal endophyte isolates tested for L-asparaginase activity (Table 3), thirty-eight isolates were positive for extracellular L-asparaginase and formation of pink zones was evident on Modified Czapex Dox (McDox) medium. The pink zone diameter varied from 15.3 to 58.2 mm (Table 3). *Fusarium proliferatum* showed maximum enzyme activity (Fig. 1), followed by *Plenodomus tracheiphilus*. All six *Fusarium* species in this study could produce L-asparaginase and the pink zones were measured above 27.3 mm. Forty-six isolates did not produce L-asparaginase, including all endophytic fungi isolated from *M. parthenium*. Also, our results demonstrated that the basidiomycetous endophytes did not produce L-asparaginase. Thirty-eight fungal strains exhibiting positive enzyme activities were selected for quantitative assay of L-asparaginase.

## Estimation of L-asparaginase production by Nesslerization

L-asparaginase activities of thirty-eight fungal endophytes were recorded to range of 0.019–0.492 unit/mL$^{-1}$ (Table 3). The isolates of *F. proliferatum* obtained from *A. altissima,* exhibited a maximum enzyme activity with 0.492 unit/mL$^{-1}$, followed by *P. tracheiphilus* isolated from *A. altissima* (0.481 unit/mL$^{-1}$). *F. oxysporum* and *Cladosporium limoniforme* exhibited moderate enzyme activity, while *Septoria tormentillae* showed the least activity with 0.019 unit/mL$^{-1}$ of enzyme (Fig. 2). Results showed that there were significant differences among the isolates at 1% (Table 3). The percentage of L-asparaginase-producing fungal endophytes was 45.2% of the total isolated endophytes (84 isolates) with 2.5%, 3.5%, 15%, 14.2%, 3.5% and 6.5% respectively for *M. chamomilla, A. triumfetii, A. altissima, A. millefolium, A. filipendulina* and *C. intybus.*

## DISCUSSION

Eighty-four fungal endophytes belonging to Ascomycota (95%) and Basidiomycota (5%) were obtained from seven medicinal plants in Iran. All species obtained in the present

**Table 3  Fungal endophytic strains from various medicinal plants and their L-asparaginase activity.** Least Significant Difference (LSD) test; The results with different superscripts were different significantly ($p < 0.01$) according to LSD test.

| Isolate code | Fungus | Host plant | GenBank accession number | Similarity (%) | Qualitative assay (mm) | L-asparaginase Enzyme in unit/mL | LSD test |
|---|---|---|---|---|---|---|---|
| Br08 | *Fusarium proliferatum* | *Anthemis altissima* | MH245099 | 100 | 58.2 | 0.492 | A |
| Br12 | *Plenodomus tracheiphilus* | *A. altissima* | MH245100 | 99 | 58.1 | 0.481 | b |
| k100 | *Torula herbarum* | *Cichorium intybus* | MH258980 | 100 | 57.1 | 0.442 | c |
| Br18 | *Fusarium avenaceum* | *A. altissima* | MH245076 | 99 | 57.2 | 0.424 | d |
| Am72 | *Fusarium oxysporum* | *Achillea millefolium* | MH259174 | 100 | 42.4 | 0.332 | e |
| Br15 | *Cladosporium limoniforme* | *A. altissima* | MH245072 | 100 | 42.1 | 0.309 | f |
| Am13 | *Fusarium redolens* | *A. millefolium* | MH259166 | 99 | 37.7 | 0.252 | g |
| Am91 | *Alternaria infectoria* | *A. millefolium* | MH259179 | 100 | 37.3 | 0.244 | h |
| AS26 | *Fusarium sp.* | *Achillea filipendulina* | MH250005 | 98 | 37.3 | 0.242 | h |
| AM55 | *Cladosporium ramotenellum* | *A. millefolium* | MH259170 | 100 | 37.2 | 0.232 | i |
| BB05 | *Chaetosphaeronema hispidulum* | *Anthemis triumfetii* | MH245081 | 100 | 36.4 | 0.224 | ij |
| Am03 | *Septoria sp.* | *A. millefolium* | MH259176 | 99 | 35.3 | 0.208 | k |
| k11 | *Alternaria embellisia* | *C. intybus* | MH258981 | 99 | 35.3 | 0.203 | l |
| BB28 | *Alternaria sp.* | *A. altissima* | MH245085 | 98 | 35.2 | 0.202 | l |
| K24 | *Plectosphaerella cucumerina* | *C. intybus* | MH258974 | 100 | 27.5 | 0.192 | m |
| Am87 | *Fusarium sp.* | *A. millefolium* | MH259177 | 98 | 27.3 | 0.187 | mn |
| BA18 | *Epicoccum nigrum* | *Matricaria chamomilla* | MH245107 | 100 | 27.1 | 0.166 | 0 |
| Br42 | *Didymella tanaceti* | *A. altissima* | MH245108 | 100 | 26.8 | 0.157 | p |
| Br09 | *Verticillium dahliae* | *A. altissima* | MH245075 | 100 | 26.4 | 0.155 | p |
| Am39 | *Paraophiobolus arundinis* | *A. millefolium* | MH259168 | 100 | 26.1 | 0.146 | q |
| Br41 | *Ulocladium consortiale* | *A. altissima* | MH245090 | 99 | 26.1 | 0.145 | q |
| Am04 | *Septoria malagutii* | *A. millefolium* | MH259172 | 100 | 26 | 0.144 | q |
| Ba24 | *Didymella tanaceti* | *M. chamomilla* | MH245097 | 99 | 26 | 0.143 | q |
| BB26 | *Stemphylium amaranthi* | *A. triumfetii* | MH245085 | 100 | 25.7 | 0.132 | r |
| Br38 | *Aspergillus calidoustus* | *A. altissima* | MH245078 | 100 | 25.7 | 0.131 | r |
| Am64 | *Nemania serpens* | *A. millefolium* | MH259183 | 100 | 25.5 | 0.125 | s |
| k29 | *Alternaria solani* | *C. intybus* | MH258977 | 99 | 25.4 | 0.123 | s |
| BA06 | *Phoma haematocycla* | *M. chamomilla* | MH245096 | 100 | 25.1 | 0.112 | t |
| As01 | *Antennariella placitae* | *A. filipendulina* | MH250008 | 100 | 24.8 | 0.108 | w |
| Am88 | *Alternaria burnsii* | *A. millefolium* | MH259181 | 99 | 24.8 | 0.107 | w |
| Am28 | *Stemphylium amaranthi* | *A. millefolium* | MH259169 | 100 | 24.7 | 0.107 | w |
| As16 | *Acremonium sclerotigenum* | *A. filipendulina* | MH250010 | 100 | 24.5 | 0.106 | w |
| Br92 | *Lewia infectoria* | *A. altissima* | MH245070 | 99 | 24.5 | 0.105 | w |
| BR25 | *Paraphoma sp.* | *A. altissima* | MH245091 | 98 | 21.4 | 0.083 | x |
| Br31 | *Sarocladium strictum* | *A. altissima* | MH245074 | 100 | 21.1 | 0.079 | x |
| K15 | *Cladosporium tenuissimum* | *C. intybus* | MH258971 | 100 | 18.2 | 0.029 | y |
| Br34 | *Stemphylium botryosum* | *A. altissima* | MH245094 | 100 | 17.9 | 0.027 | y |
| Am51 | *Septoria tormentillae* | *A. millefolium* | MH259171 | 99 | 15.3 | 0.019 | z |

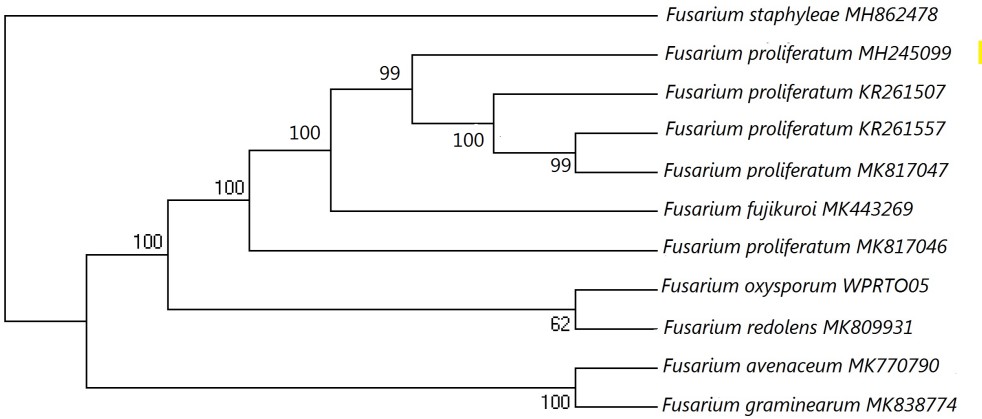

**Figure 1** **A maximum parsimony phylogeny for *Fusarium proliferatum* from ITS (Internal transcribed spacer).** Phylogenetic position of isolate MH245099 was highlighted. Bootstrap tests were performed with 1,000 replications. *Fusarium staphyleae* (MH862478) was used as an outgroup.

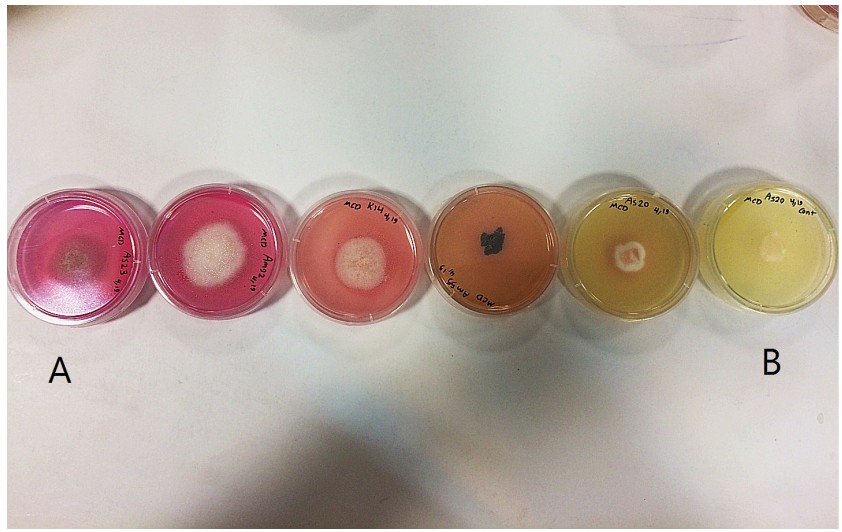

**Figure 2** **L-asparaginase activity detected by plate assay.** Colour change in the medium (yellow to pink) around colony indicates production of enzyme. (A) Isolates showing high production of L-asparaginase; (B) non-producer isolates.

study are reported for the first time as endophytes from *M. chamomilla, M. parthenium, A. triumfetii, A. altissima, A. millefolium, A. filipendulina* and *C. intybus*. In recent years, most records of fungal endophytes are Ascomycota (*Carroll, 1988; Rodrigues, 1994; Gonthier, Gennaro & Nicolotti, 2006; Arnold, 2007*) with a few species of Basidiomycota (*Petrini, 1986; Chapela & Boddy, 1988; Oses et al., 2006; Sánchez Márquez, Bills & Zabalgogeazcoa, 2007*). Endophytic Basidiomycota such as *T. versicolor, B. adusta, T. biforme* and *S. commune* were isolated from medicinal plants during this study are white- rot fungi. This finding is consistent with research that showed that most basidiomycetous endophytes, although not

pathogenic in some hosts, incite white-rot in decaying wood (*Oses et al., 2006*; *Thomas et al., 2008*).

The frequency of colonization of endophytic fungi was higher in the stem compared to the other plant tissues. Previous studies have also reported high colonization frequency of endophytic fungi in stem tissues (*Bezerra et al., 2015*). *Bezerra et al. (2015)* extrapolated that highest frequency of colonization in the stem may be due to spore abundance of a few dominant endophytes in stem tissue. Similarly, *Verma et al. (2013)* demonstrated that the diversity of endophytic fungi was highest in the stem. The diversity and frequency of colonization of fungal endophytes are influenced by the host tissue (*Rodrigues, 1994*) and environmental factors (*Clay, 1986*). However, most studies reported that leaf tissues yield a higher diversity of endophytes (*Verma et al., 2007*; *Gond et al., 2012*).

In our study, most of the fungal species (e.g., *T. versicolor*) were isolated from a single host. In contrast, few fungal species, such as *A. sclerotigenum,* were common in multiple host plants. Endophytic assemblages tend to be distributed in specific hosts and specific tissues (*Siqueira et al., 2011*; *Xing, Guo & Fu, 2010*). Some dominant endophytes have been recovered from every part of plants such as *F. avenaceum.* This may be due to the ability of endophytes to penetrate from one part of plant to another (*Manasa & Nalini, 2014*).

L-asparaginase is one of the most effective antineoplastic agents for the treatment of acute leukemia (*Nakamura, Wilkinson & Woodruff, 1999*). It is produced by plants and a variety of microbial sources including fungi (*Serquis & Oliveira, 2004*). Endophytes from medicinal plants are rich sources of novel compounds (*Hwang et al., 2011*). In this study, L-asparaginase activities of the fungal endophytes from different medicinal plant species were evaluated. Eighty-four fungal endophytes were examined for the L-asparaginase activity. Thirty-eight of these demonstrated the ability to metabolize L-asparagine. The fungi that were good producers of this enzyme belonged to the genus *Fusarium,* followed by species of *Alternaria* and *Cladosporium.* Furthermore, these species has been reported to produce L-asparaginase (*Serquis & Oliveira, 2004*; *Theantana, Hyde & Lumyong, 2009*). L-asparaginase activity was not observed in the endophytes from *M. parthenium.* This may be due to the low diversity of endophytes that were obtained from this medicinal plant. Although, the *S. tormentillae* showed pink zones in the agar assay, enzymatic activity was low based on further quantitative analysis. The reason for the absence of enzyme activity in the quantitative estimation may be attributed to differences in the ability of the fungi to produce the enzyme in solid and liquid states (*Holker, Hofer & Lenz, 2004*). According to available literature, this is the first record of L-asparaginase production by endophytic fungi of the host plants examined in the present study.

## CONCLUSIONS

Studies here revealed that the diversity of some endophytic fungal communities was influenced by host plants and tissues. We isolated numerous fungal endophytes from seven healthy medicinal plants of Iran. Endophytes that were able to produce L-asparaginase belonged to the genera *Plectosphaerella, Fusarium, Stemphylium, Septoria, Alternaria, Didymella, Phoma, Chaetosphaeronema, Sarocladium, Nemania, Epicoccum, Ulocladium*

*and Cladosporium.* An isolate of *Fusarium proliferatum* was found to have the highest L-asparaginase enzyme activity. Our findings are consistent with the hypothesis that endophytes associated with medicinal plants have potential medicinal properties. We found that production of L-asparaginase by endophytic fungi may provide an alternative source for this enzyme. Further studies involving enzyme isolation are necessary to prove the utility of the L-asparaginases derived from fungal endophytes.

### Funding

This work was supported by grants to the first author Sareh Hatamzadeh from the postgraduate committee of the Research Vice President of Gorgan university of Agricultural Sciences and Natural Resources, Iran. This work was also supported by the New Jersey Agricultural Experiment Station and Multistate (Project number 4147), United States of America. The funders had no role in study design, data collection and analysis, decision to publish, or preparation of the manuscript.

### Grant Disclosures

The following grant information was disclosed by the authors:
Postgraduate committee of the Research Vice President of Gorgan university of Agricultural Sciences and Natural Resources, Iran.
New Jersey Agricultural Experiment Station and Multistate (Project number 4147), United States of America.

### Competing Interests

The authors declare there are no competing interests.

### Author Contributions

- Sareh Hatamzadeh conceived and designed the experiments, performed the experiments, analyzed the data, prepared figures and/or tables, authored or reviewed drafts of the paper, and approved the final draft.
- Kamran Rahnama conceived and designed the experiments, analyzed the data, authored or reviewed drafts of the paper, and approved the final draft.
- Saeed Nasrollahnejad analyzed the data, authored or reviewed drafts of the paper, and approved the final draft.
- Khalil Berdi Fotouhifar, Khodayar Hemmati and Fakhtak Taliei performed the experiments, authored or reviewed drafts of the paper, and approved the final draft.
- James F. White conceived and designed the experiments, authored or reviewed drafts of the paper, and approved the final draft.

### Data Availability

The raw data are available as a Supplemental File.

## Supplemental Information

Supplemental information for this article can be found online at http://dx.doi.org/10.7717/peerj.8309#supplemental-information.

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
