# Peer review of "Isolation and identification of L-asparaginase-producing endophytic fungi from the Asteraceae family plant species of Iran"

_PeerJ, doi:10.7717/peerj.8309_

## Round 0.1 · original submission · Major Revisions

We received two reviews with remarks in PDF files. The lists of remarks are long. Though the remarks are rather technical. I believe you can update the text soon.

·

Basic reporting

No comments

Experimental design

No comments

Validity of the findings

No comments

Additional comments

I have reviewed the manuscript by Hatamzadeh et al. The premise behind the manuscript is nice, however, some points need to be addressed. The main revisions which need to be made and other queries are listed below. Other minor issues, which the authors could utilize to improve the manuscript, are indicated on the annotated manuscript which I have attached to this review.

Comments to be addressed:
What was the control while isolating fungal endophytes?
Have authors submitted a voucher specimen(s) of L-asparaginase-producing endophytic fungi in any repository?
Line 137: Identification of Fungal Endophytes “No results have been presented regarding the identification of endophytic fungi in the text” What was identity % of the isolated strains with NCBI strain(s)? It should be mentioned in the text.
The discussion needs to be strengthened.
Table 3: Nothing has been described about the data presented in table 3. Please elaborate.

·

Basic reporting

The authors described the potential of endophytic fungi from seven medicinal plants to produce L-Asparaginase, an enzyme used for industrial and pharmaceutical purposes, particularly as anticancer agents. It is obvious while reading the manuscript that an intensive work was done by the authors. However, many points in the manuscript need to be carefully addressed to improve not only the quality of the research but also its scientific value (check the main manuscript for more comments)
1. Overall, the English language needs to be significantly improved.
2. Abstract
The abstract is not a reflexion of the manuscript, please this need to be carefully addressed. For instance while reading the abstract you may think that only 104 isolates were obtained but that is not the case in the main document. Another point, isolates were first screened using a qualitative method before quantification. There is no such indication in the abstract. For the form, there are too many repeated sentences in the abstract; some data are missing, etc.. (See the reviewed manuscript version for more comments).
3. Introduction
In the introduction, please give the appropriate definition of endophytes. Many references are not appropriately cited (this is valuable for other parts of the document), while many claims do not have references. This can be seen as plagiarism. Please this issue needs to be carefully addressed. The introduction have some repeated sentences as well. More importantly, the rationale for investigating these plants species is not mentioned (more comments in the main manuscript).

4. Results
Identification of fungi: a proper description of the colonization frequency is needed. In fact, the experiment was performed and data collected so the presentation of results should be as clear as possible for better appreciation of the quality and amount of work conducted.

The activity was measured qualitatively by measuring diameter on agar plates. There is a need here to show the range of diameters obtained and the most potent according to the method used. Thereafter, the criteria used to select those used in Nessterization assay should be mentioned to create fluidity in the text. Moreover, how many endophytes were screened qualitatively? How many showed potency?? What was the higher diameter of production???? Many questions need to be addressed in this section.
Overall, the presentation of data needs careful reconsideration. Details information are needed to better appreciate the scientific values.
5. Discussion
In the discussion section only six plants are mentioned and while reading the results section, there is no indication of the inactivity of endophytes from some plants and not others. This highlights the lack of clarity in the writing. Issue to be addressed.
The misuse of references is noticeable in this section as well as redundant sentences.
The most noticeable problem in this section is the fact that no discussion is provided on the isolation frequency, endophytes colonization per plants, the dominant species etc. Overall, many questions need clear answers. For instance, which plant had the most endophytes colonization? which organ per plant was the most colonized? in general which organs, why? could the environment, time of sampling had any influence on that colonization frequency? ........
6. Conclusion
No indication from this study revealed that endophytes from Asteraceae produced L-asparaginase with superior activity, since no activity was actually tested.
The authors projected to perform the clinical trial on pure asparaginase, and not the actual preclinical study. In general, the conclusion needs carefully editing.

Experimental design

The protocol of isolation needs to be carefully amended (my comments are included in the manuscript)

Validity of the findings

Acceptable

Additional comments

I encourage the authors to revise their manuscript following the comments and appended edits in the main text.

---

## Round 0.2 · Major Revisions

The reviewers have some critical comments on the manuscript. However, they were so kind to provide remarks in the separate file. Please take it into account in the revision. Please pay attention to English presentation of the text.

·

Basic reporting

I have reviewed the manuscript by Hatamzadeh et al. The manuscript still needs a major revision. I would suggest that authors should carefully read what they are writing as there are a lot of typographical mistakes. A lot of information is missing in the material and methods section. A sound rationale for their study is still needed. A proper explanation is needed in some parts of the discussion and the conclusion needs to be strengthened. Authors must recheck the results they have presented particularly in Table 3. The manuscript still needs attention on the English at some places. I would like to suggest some improvements to make the manuscript more clear and focused. The other issues, which the authors could utilize to improve the manuscript, are indicated on the attached file.

Experimental design

The authors need to focus on the material and methods section as a lot of information and details are missing.

Validity of the findings

The authors need to validate some results which I have pointed in the attached file.

·

Basic reporting

This manuscript describes isolation, identification and quantification of L-asparaginase activity of the endophytic fungi isolated from seven Iranian medicinal plants. The topic is generally interesting. The methods used seem appropriate and the research ended with novel conclusion. However, the manuscript needs revision for better clarity, pointing out the novelty and coherence.

Experimental design

The experimental design seems okay but this section needs enough technical details for reproduction by the other researchers.

Validity of the findings

The findings seem valid. However, the description of novelty and perspection in the discussion section is inadequate with relevant recent literature.

Additional comments

This manuscript needs major revision including editing by a native English speaker before considering for publication. Scientific names in the Tables should be critically checked and must follow the standard style (first appearance full name and then abbreviated form for the generic part). I made some comments and also edited in the manuscript.

---

## Round 0.3 · accepted · Accept

As I see all the detailed comments from the reviews were taken into account. The manuscript got enough English editing for the publication. I recommend publish it in the current form

I'd only recommend make more precise title - not just 'some medical plant species..' but 'Asteraceae family plant species.."

or 'seven plant species..' Just suggestion for the final version.

·

Basic reporting

The authors substantially improved their manuscript although a slight grammatical errors are still remained which I think would be removed during editorial work.

Experimental design

Seems sound.

Validity of the findings

Yes, the data seem original and valid.

Additional comments

Total number of references are too high. Please delete some references that are less relevant to the article.